# Measuring recent effective gene flow among large populations in *Pinus sylvestris*: Local pollen shedding does not preclude substantial long-distance pollen immigration

Azucena Jiménez-Ramírez[1,2]*, Delphine Grivet[1], Juan José Robledo-Arnuncio[1]*

**1** Department of Forest Ecology & Genetics, Forest Research Center (INIA, CSIC), Madrid, Spain,
**2** Department of Genetics, Faculty of Biological Sciences, Complutense University of Madrid, Madrid, Spain

* jjrobledo@gmail.com (JJRA); jimenez.azucena@inia.es (AJR)

**Data Availability Statement:** All relevant data are within the manuscript and its Supporting Information files.

## Abstract

The estimation of recent gene flow rates among vast and often weakly genetically differentiated tree populations remains a great challenge. Yet, empirical information would help understanding the interaction between gene flow and local adaptation in present-day non-equilibrium forests. We investigate here recent gene flow rates between two large native Scots pine (*Pinus sylvestris* L.) populations in central Iberian Peninsula (Spain), which grow on contrasting edaphic conditions six kilometers apart from each other and show substantial quantitative trait divergence in common garden experiments. Using a sample of 1,200 adult and offspring chloroplast-microsatellite haplotypes and a Bayesian inference model, we estimated substantial male gametic gene flow rates (8 and 21%) between the two natural populations, and even greater estimated immigration rates (42 and 64%) from nearby plantations into the two natural populations. Our results suggest that local pollen shedding within large tree populations does not preclude long-distance pollen immigration from large external sources, supporting the role of gene flow as a homogenizing evolutionary force contributing to low molecular genetic differentiation among populations of widely distributed wind-pollinated species. Our results also indicate the high potential for reproductive connectivity in large fragmented populations of wind-pollinated trees, and draw attention to a potential scenario of adaptive genetic divergence in quantitative traits under high gene flow.

## Introduction

Gene flow, the successful transfer of genes among populations, determines not only the genetic connectivity and structure of metapopulations, but also the degree of local adaptation, which results from the balance between gene flow, divergent selection among populations in diverse environments, and random genetic drift within populations [1]. Gene flow is known to vary considerably among species, populations and individuals, as well as over time [2, 3]. Understanding the relative contributions of gene flow, genetic drift and natural selection to

**Funding:** This work was supported by CGL2015-64164-R project grant to JJRA and DG and by BES-2016-078969 PhD grant to AJR, both co-financed by MINECO (https://www.mineco.gob.es) and ERDF (https://ec.europa.eu/regional_policy/en/funding/erdf/). Research was also supported by AEG 17-048 project grant from MAPAMA (https://www.mapama.gob.es), established in the frame of the measure 15.2 "support to the conservation and use of forest genetic resources" and under Regulation (EU) No 1305/2013 of the European Parliament and of the Council of 17 December 2013 on support for rural development by the European Agricultural Fund for Rural Development (EAFRD; https://ec.europa.eu/info/food-farming-fisheries/key-policies/common-agricultural-policy/rural-development_en) with 75% co-financing. The funders had no role in study design, data collection and analysis, decision to publish, or preparation of the manuscript. No additional external funding was secured for this study.

**Competing interests:** The authors have declared that no competing interests exist.

geographical patterns of adaptive genetic variation has been a recurring theme of evolutionary biology. Whereas genetic drift generally reduces the efficiency of natural selection, especially in small populations [4], gene flow may interact in contrasting ways with local adaptive processes. On the one hand, it is thought to hamper local adaptation in heterogeneous environments by disrupting co-adapted gene complexes and changing allele frequencies in a direction opposite to divergent natural selection ('migration load' effect; [5]). On the other hand, gene flow can reduce the deleterious effects of inbreeding and increase the genotypic variance available for selection, facilitating local adaptation ('genetic rescue' effect; [6]). The relative importance of these effects will vary, depending on the precise level of gene flow among populations, the strength of divergent selection, and the effective size of the populations. Negative adaptive effects are expected to increase for high levels of gene flow across heterogeneous environments, which may ultimately lead to the eventual displacement of locally adapted genotypes ('gene swamping'; [5]). Lower levels of gene flow may by contrast help natural populations to adapt to their local environment by increasing the genetic variance available for selection and reducing inbreeding.

Assessing the spatial scale and the magnitude of gene exchange among populations is therefore essential to understand how it affects adaptive dynamics within species. Quantifying gene flow patterns will be especially relevant for widely-distributed species with potential for long-distance dispersal across heterogeneous environments, to which they may be locally adapted. This is likely the case of many widespread wind-pollinated tree species of *Pinus*, *Picea*, *Betula*, *Quercus* and others. In particular, the potential for gene exchange among wind-pollinated conifer populations is high, as they are well known for long-distance pollen transport over hundreds of kilometers [1]. However, direct evidences of effective long-distance pollen transport resulting in actual gene flow events are still scarce [7–9]. Although observed background pollination from unknown sources into isolated stands is likely to be the result of long distance dispersal events (e.g. [10]), experimental constraints have prevented precise measurements of the actual scale of this potentially long-distance pollen immigration process. Maximum documented dispersal distances are typically orders of magnitude lower for effective pollination than for mere pollen transport events [1]. It is unclear whether discrepancies between the documented scales of potential versus effective pollen dispersal largely derive from obstacles to successful long-distance mating, such as phenological asynchronies [11] and competition with local pollen [12] or, rather, from greater difficulties in assessing effective dispersal over broad areas [8].

Broad-scale quantification of pollen dispersal patterns is indeed notoriously difficult. Since early experimental attempts to estimate contemporary pollen transport distances in tree populations using traps or dyes [13], a number of analytical procedures have been developed to make best use of effective pollen dispersal information derived from molecular marker assays (reviewed in [14]). Among them, those based on paternity analysis have been widely used to characterize spatial patterns of pollen dispersal in tree species, though mostly within populations [15]. Among the few formal procedures available for estimating contemporary effective pollen flow over broad spatial scales, genetic parentage analysis and genetic assignment methods are the most commonly employed. Genetic parentage exclusion methods have been developed for assessing immigration into small populations, but they are not feasible for large and dense stands, because they require exhaustive genotyping of all potential parents within the recipient population [16]. Genetic assignment methods allow the establishment of the population origin of every individual in a sample without exhaustive genotyping of all individuals from candidate source populations, enlarging thus the scale of analysis [17]. Basic formulations focus on individual assignment tests and are inefficient for unbiased estimation of recent gene flow rates, but several developments have targeted migrant proportions instead, providing

potentially accurate estimates of recent gene flow rates [18, 19], and specifically of seed and pollen gene flow rates among plant populations [20, 21]. The power of genetic assignment-based methods is substantially decreased when genetic differentiation among source populations is modest, however, in which case it becomes especially advisable to evaluate numerically expected estimation errors [19].

The estimation of recent gene flow rates among vast and often weakly genetically differentiated tree populations remains a great challenge, even if it would provide necessary empirical information about the poorly understood interaction between gene flow and local adaptation in present-day non-equilibrium forests. Along this line of work, the objective of this study is to estimate recent gene flow rates between two large natural Scots pine (*Pinus sylvestris* L.) populations in central Iberian Peninsula (Spain). Even if they are separated by only 6 km, well within the range of pollen transport distances for this species, they intriguingly show substantial genetic divergence in quantitative traits [22, 23]. We hypothesize that there must have been high recent gene flow among the two pine populations, given the high pollen movement levels reported in this species and the typically low reproductive barriers among conspecific pine populations [24]. Failing to reject this working hypothesis would suggest that the well-known long-distance pollen transport in wind-pollinated trees can actually result in effective pollen migration even into large stands with abundant local pollen production. Moreover, it might suggest that very strong divergent selection must be operating in our study system, in order to counteract the homogenizing effect of high among-population gene flow rates, which would make this system a good candidate for further studies of local adaptation. To achieve our goal, we measure recent among-population gene flow rates using uniparentally inherited markers and a Bayesian approach that accounts for haplotype frequency estimation uncertainty.

## Materials and methods

### Ethics statement

The Dirección Provincial de Agricultura, Medio Ambiente y Desarrollo Rural en Guadalajara (Dirección General de Política Forestal y Espacios Naturales) authorized the field work conducted in this study (permit number 1080766).

### Species, study sites and field sampling

Scots pine is an ecologically and economically important species widely distributed throughout Eurasia, with a continuous distribution in the north and more fragmented populations in the southern margins [25]. The species is tolerant to poor soils, drought and frost and is found in contrasting climatic conditions and ecological habitats. It is monoecious, predominantly out-crossing, anemochorous and wind-pollinated, with potential for very long-distance pollen dispersal [26–28]. The geographic distribution of molecular genetic diversity in Scots pine reflects the dispersal biology of the species, with low overall genetic differentiation and geographic structure in the more continuous part of the distribution, yet with subtle geographic structure at the range margins and even in central distribution areas [26]. Contrastingly, many putatively adaptive quantitative traits show substantial spatial genetic variation across the species range, consistent with local adaptation to environmental gradients [26]. The species reaches its southernmost distribution limit in the Iberian Peninsula, where populations scattered on different mountain chains show evidences of local adaptation to regional habitat heterogeneity [29, 30].

We focused here on two natural Scots pine populations in Sierra de Ayllón, central Spain, one in Campisábalos (41˚14'34.7"N, 3˚08'51.3"W) and the other in Galve de Sorbe (41˚

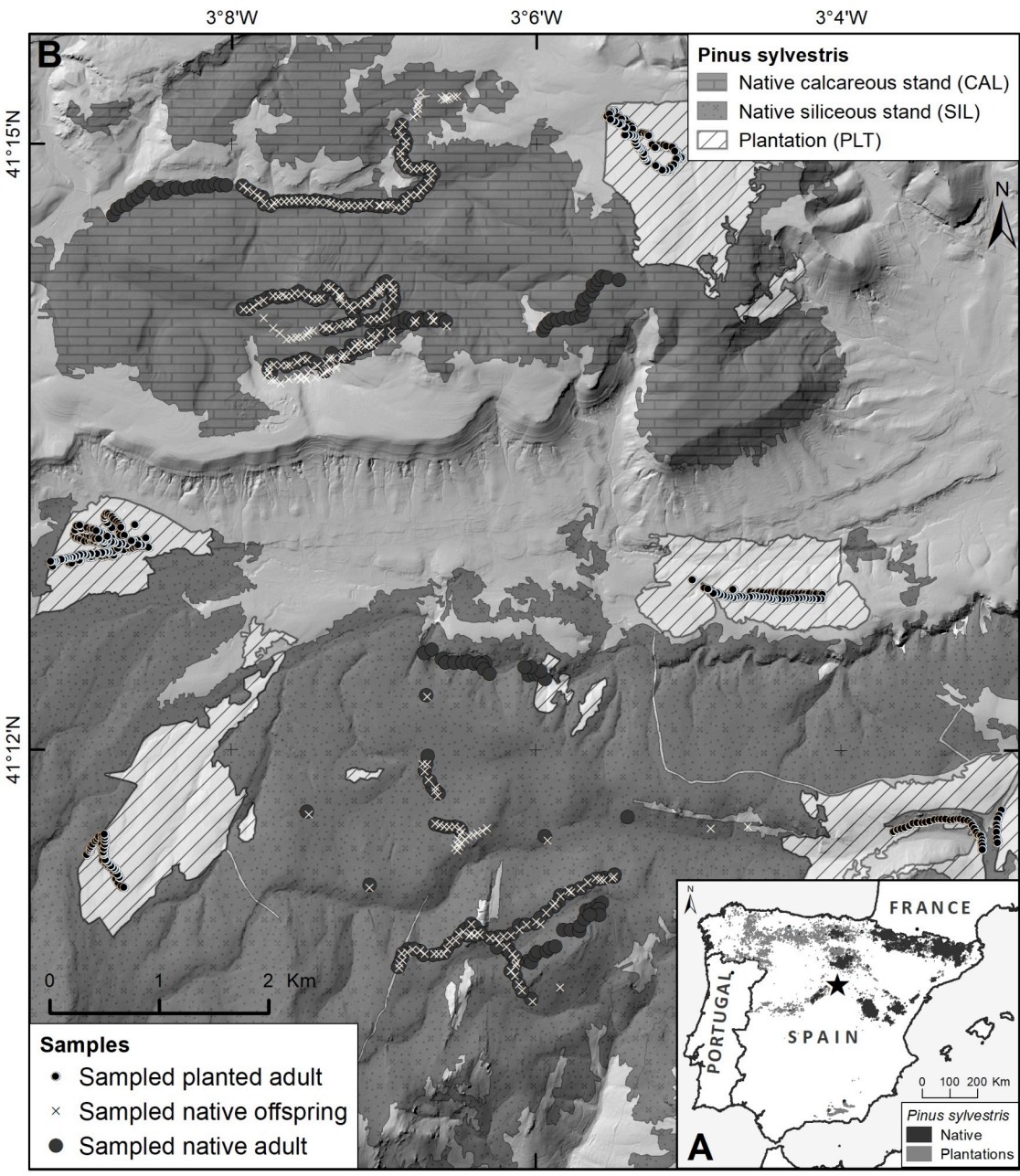

**Fig 1. Maps of *Pinus sylvestris* study site in Sierra de Ayllón (Spain).** (A) Location of Sierra de Ayllón (marked with a star) and distribution of native and planted populations of the species within the Iberian Peninsula. (B) Distribution of sampled native and planted adult and offspring samples within the calcareous and siliceous populations. BDLJE CC-BY 4.0 ign.es, MDT05 2015 CC-BY 4.0 scne.es, MFE50_19 2003 miteco.gob.es.

12'25.6"N, 3°06'45.8"W) (Fig 1). The two populations grow in contrasting edaphic conditions: while the one at Campisábalos (code CAL) thrives on calcareous-dolomitic soils, the Galve de Sorbe one (SIL) is distributed on siliceous slate-quartzite soils. Despite being at relatively close distance (about 6 km away), allowing potentially high levels of gene flow, they show substantial quantitative genetic differentiation in early survival and in early phenological and growth traits at provenance trials [22, 29]. Due to their ecological and genetic differentiation, they were

classified as separate seed sources by the Spanish government. Several scattered planted stands (PLT) of unknown, but presumably local, origin were introduced in the second half of the 20-th century around and within the two natural stands (Fig 1). Being over half a century old, the plantations have likely reached full reproductive competence.

In order to estimate reference paternal haplotypic frequencies for gene flow inference, we sampled adult individuals along transects across the two natural stands and the nearby plantations (Fig 1). Specifically, we collected needle tissue from a total of 800 adult trees: 200 from CAL and SIL, respectively, and 400 from the plantations, keeping a minimum separation distance of 30 m between individual trees to spread samples and avoid potential individual relatedness. The goal was to estimate male gametic gene flow among naturally established recruits under the canopy of the natural stands, for which we additionally sampled 400 young individuals (hereinafter referred to as "offspring") recruited in scattered patches along the same adult-sampling transects within the two natural populations, 200 offspring from CAL and SIL, respectively. Sampled offspring varied in height from 0.15 to 1.5 m, with roughly estimated ages from a few years to a couple of decades. Our gene flow estimates will thus represent an average over approximately the last twenty years. The total 1 200 samples (800 adults and 400 offspring) were placed individually in paper envelopes, dried with silica gel, and stored at room temperature in the laboratory until DNA extraction.

## Laboratory analysis

We extracted DNA from 20 mg of dried needle tissue from each of the 1 200 samples, using Invisorb DNA Plant HTS 96 Kit/C (INVITEK Molecular GmbH, Berlin, Germany) and the concentrations of all DNA solutions were determined using a Qubit 3.0 fluorometer (Life Technologies, California, USA). All the samples were genotyped at six chloroplast microsatellites (cpSSRs): Pt15169, Pt26081, Pt30204, Pt36480, Pt71936 and Pt87268 [31], following the polymerase chain reaction (PCR) amplification protocols described in S1 Appendix. Amplified fragments were separated using an ABI 3730 genetic Analyzer (Applied Biosystems, Carlsbad, CA) at the Genomics and Proteomics Service of the Universidad Complutense de Madrid, and fragment sizes were determined using the GeneScan™ 500 LIZ® Size Standard (Applied Biosystems) with the software GENEMAPPER ver. 3.0 (Applied Biosystems).

## Analysis of genetic diversity and differentiation

Haplotypes were defined as unique combinations of size variants across the six used cpSSR. For each group of adult and offspring samples, we computed the observed number of haplotypes ($nh$), the observed number of haplotypes standardized to the minimum observed sample size across samples ($nh^*$; [32]), and the effective number of haplotypes ($nh_e$; using eqn 9 in [33]). Standard errors for $nh_e$ were computed by bootstrapping individuals within samples 1 000 times, and differences in $nh_e$ between samples were compared via bootstrap-based two-sample tests [34]. We estimated genetic divergence between groups using the AMOVA-based $F_{ST}$-statistic, implemented in ARLEQUIN ver. 3.5 software, and Jost's differentiation index ($D_{est}$; [35]), as implemented in DEMEtics ver. 0.8–5 [36].

## Model for estimating gene flow rates

Our goal is to estimate the proportion of male gametes originating from each of $J$ populations among different offspring cohorts of a subset $I$ of the same populations (in our study case $J = 3$ and $I = 2$; in general $1 \leq I \leq J$). We denote $\mathbf{m} = \{m_{ij}\}$ the matrix of size $I^*J$ where $m_{ij}$ is the proportion of offspring from population $i$ sired by fathers from population $j$. A set of $\{A_1, A_2, \ldots, A_J\}$ adult individuals are randomly sampled from the $J$ candidate source populations, and a set

of $\{O_1, O_2, \ldots, O_I\}$ offspring from the *I* recipient populations of interest. We assume that all possible source populations of male gametes are sampled. Both the adult and offspring samples are genotyped using the same array of haploid paternally-inherited markers, yielding a total of *K* different haplotypes observed across the entire collection of genotyped adult and offspring individuals. Let $\mathbf{N} = \{N_{ih}\}$ and $\mathbf{n} = \{n_{ih}\}$ be the matrices of adult and offspring haplotypic counts, where $N_{ih}$ and $n_{ih}$ are the observed numbers of copies of haplotype *h* among, respectively, the adult and offspring samples collected from population *i*. Note that $\sum_h N_{ih} = A_i$ and $\sum_h n_{ih} = O_i$. The unknown adult population haplotypic frequencies are given by a matrix $\mathbf{p} = \{p_{ih}\}$, giving the frequency of haplotype *h* in population *i*.

Assuming that pollen fecundity is haplotype independent, we can write the probability of observing the *h*-th haplotype in the offspring sample of population *i*, given the unknown migrant proportions $\mathbf{m}$ and population haplotypic frequencies $\mathbf{p}$, as

$$\Pr(H_{ih}|\mathbf{m}, \mathbf{p}) = \sum_{j=1}^{J} m_{ij} p_{jh}, \qquad \text{Eq 1}$$

from which we can compute the likelihood of the full set of observed offspring haplotypic counts $\mathbf{n}$ across recipient populations:

$$\Pr(\mathbf{n}|\mathbf{m}, \mathbf{p}) = \prod_{i=1}^{I} \prod_{h=1}^{K} \left[\Pr(H_{ih}|\mathbf{m}, \mathbf{p})\right]^{n_{ih}}. \qquad \text{Eq 2}$$

On the other hand, the likelihood of the observed adult haplotypic counts under random sampling is given by the joint probability of independent multinomial draws across populations:

$$\Pr(\mathbf{N}|\mathbf{p}) = \prod_{j=1}^{J} \text{Mult}(\mathbf{N}_j, \mathbf{p}_j), \qquad \text{Eq 3}$$

where $\text{Mult}(\mathbf{N}_j, \mathbf{p}_j)$ is the multinomial probability of $\mathbf{N}_j = \{N_{j1}, N_{j2}, \ldots, N_{jK}\}$ draws from *K* classes with probabilities $\mathbf{p}_j = \{p_{j1}, p_{j2}, \ldots, p_{jK}\}$. In the case of our empirical data, note that by using the same haplotypic frequencies in Eqs 2 and 3 we are implicitly assuming that our contemporary adult sample provides a good approximation of the haplotypic frequencies of the fathers that have sired offspring established during the last couple of decades. This should be a good approximation, given that the studied Scots pine populations are large, that they are naturally regenerated only every 100–120 years, and that there had not been important recent cuttings in the study area.

We assumed flat prior distributions (*f*) for both migrant proportions and population haplotypic frequencies. In particular, the proportions of offspring from the *i*-th population sired by males from different sources ($\mathbf{m}_i$) were assumed to follow a flat Dirichlet prior, i.e. symmetric with parameter equal to one, $\mathbf{m}_i \sim \text{Dir}(\alpha = 1)$. Similarly for the frequency of the *K* haplotypes at each population *i*, we assumed $\mathbf{p}_i \sim \text{Dir}(\alpha = 1)$. These priors imply that any allele frequency (or migration rate) distribution is equally likely a priori, including those with some (up to all but one) classes with zero frequency.

Given the vectors of haplotypic data $\mathbf{n}$ and $\mathbf{N}$, the joint posterior distribution of $\mathbf{m}$ and $\mathbf{p}$ is given by Bayes' rule:

$$F(\mathbf{m}, \mathbf{p}|\mathbf{n}, \mathbf{N}) \propto \Pr(\mathbf{n}|\mathbf{m}, \mathbf{p})\Pr(\mathbf{N}|\mathbf{p})f(\mathbf{m})f(\mathbf{p}), \qquad \text{Eq 4}$$

where the *f* functions are the prior distributions of individual parameters. We used the MCMC algorithm described in S2 Appendix to estimate the joint posterior distribution of Eq 4. Point

estimates, and 95% credibility intervals (CI), of parameters were then obtained as the median, and as the 0.025 and 0.975 percentiles, of the posterior density, respectively. Note that, unlike our previous maximum-likelihood model to infer contemporary male gametic immigration rates [8], the present method jointly estimates population haplotypic frequencies, which tends to reduce biases in migration estimates. Maximum-likelihood approaches could also be potentially extended to jointly estimate haplotypic frequencies, e.g. using the expectation-maximization algorithm, but they should provide virtually identical estimates to our Bayesian model with flat priors, while being less flexible to accommodate more complex inference models or eventually available prior information.

## Simulation analysis of model performance

We used Monte Carlo simulations to calculate the expected bias, root mean square error (RMSE) and CI non-coverage rate of **m** estimates obtained with Eq 4. We did so under contrasting levels of assumed gene flow and for different assumed population haplotypic frequency distributions, given the actual number of populations, sample sizes and observed haplotypic counts. In particular, we investigated the effect on estimates of **m** of increasing values of simulated $m_{ij}$, ranging from zero (no migration) to 0.6, and including scenarios with equal or unequal migration rates from the different external sources (Table 1). We also considered one last scenario with simulated migration rates equal to the estimated ones (see Table 1 and Results). We tried to simulate populations approximating the actual population genetic structure of the real ones, while accounting for potential biases resulting from undetected low-frequency haplotypes in field samples. For this purpose, simulated adult population haplotypic frequencies were set at the posterior frequency, given the empirically observed haplotypic counts, under the conservative prior assumption that all the populations have an identical set of *nh* equifrequent haplotypes [37]. We considered three alternative values of *nh* in the simulations: either equal to, twice as large as, or four times as large as, the total number of observed haplotypes across all adult and offspring samples. Once the assumed population haplotypic frequencies were used to simulate adult and offspring samples, they were subsequently considered to be unknown and were estimated from the simulated samples during inference, reflecting thus errors owing to small adult and offspring samples. See S3 Appendix for details

**Table 1. Assumed male gametic gene flow rates in numerical simulations used to assess the inference model performance.**

| Scenario | $m_{CC}$ | $m_{CS}$ | $m_{CP}$ | $m_{SC}$ | $m_{SS}$ | $m_{SP}$ |
|---|---|---|---|---|---|---|
| A | 1 | 0 | 0 | 0 | 1 | 0 |
| B | 0.95 | 0.05 | 0 | 0.05 | 0.95 | 0 |
| C | 0.95 | 0 | 0.05 | 0 | 0.95 | 0.05 |
| D | 0.9 | 0.05 | 0.05 | 0.05 | 0.9 | 0.05 |
| E | 0.7 | 0.3 | 0 | 0.3 | 0.7 | 0 |
| F | 0.7 | 0 | 0.3 | 0 | 0.7 | 0.3 |
| G | 0.4 | 0.3 | 0.3 | 0.3 | 0.4 | 0.3 |
| H | 0.4 | 0.6 | 0 | 0.6 | 0.4 | 0 |
| I | 0.4 | 0 | 0.6 | 0 | 0.4 | 0.6 |
| J | 0.4 | 0.1 | 0.5 | 0.1 | 0.4 | 0.5 |
| K | 0.4 | 0.5 | 0.1 | 0.5 | 0.4 | 0.1 |
| L | 0.25 | 0.08 | 0.64 | 0.21 | 0.34 | 0.42 |

Each simulated scenarios assumes a different set of six pairwise male gametic gene flow rates (*m*), with the first and second subindexes of *m* indicating the recipient and source population, respectively, with codes *C* (calcareous natural population), *S* (siliceous natural population) and *P* (plantations).

**Table 2. Genetic diversity estimates for *Pinus sylvestris* adult and offspring samples from calcareous and siliceous soils at Sierra de Ayllón.**

| Sample | $n$ | $nh$ | $nh^*$ | $nh_e$ (SD) |
|--------|-----|------|--------|-------------|
| CAL-A | 196 | 98 | 95.6 | 66.4 (12.6) |
| SIL-A | 198 | 97 | 94.2 | 81.3 (12.9) |
| PLT-A | 397 | 148 | 95.7 | 80.4 (8.0) |
| CAL-O | 196 | 93 | 90.8 | 75.2 (12.1) |
| SIL-O | 188 | 98 | 98.0 | 82.9 (16.3) |

CAL-A, SIL-A and PLT-A, adult samples from calcareous and siliceous natural populations and from nearby plantations, respectively; CAL-O and SIL-O, offspring samples from calcareous and siliceous natural populations; $n$, number of successfully genotyped samples; $nh$, observed number of chloroplast haplotypes; $nh^*$, observed number of chloroplast haplotypes standardized to the minimum sample size of 188; $nh_e$ (SD), effective number of chloroplast haplotypes (standard deviation).

about the used Monte Carlo simulations. All inference and simulation algorithms were coded in C++.

## Results

### Genetic diversity and population differentiation

Estimated haplotypic genetic diversity was similarly high across calcareous and siliceous natural stands and the plantations, as well as across adult and offspring cohorts within natural stands (Table 2). Although the larger sample size of the plantations resulted in a substantially higher observed number of haplotypes than in the rest of samples ($nh = 148$ versus $93-98$), the difference largely disappeared after standardizing estimates to the minimum sample size ($nh^* = 91-98$ across all samples). Estimates of the effective number of haplotypes ranged from a low of 66 (for the calcareous natural stand) to a high of 83 (among offspring of the siliceous stand), but pairwise differences were not statistically significant (bootstrap two-sample tests, $P > 0.1$).

Genetic differentiation estimates were similar between natural adult stands on different soils, between natural and planted stands, and between adult and offspring cohorts within natural stands (Table 3). In particular, $F_{ST}$-estimates were very small ($\leq 0.001$) and not significantly different from zero for any sample pair, constrained by high within-population haplotypic diversity. Estimates of Jost's $D_{est}$, which reflect dissimilarity in population haplotypic frequencies independently of within-population diversity, were larger, ranging from 0.124 to 0.238, but because of their large variance only the maximum estimate (between SIL-A and CAL-O) was significantly different from zero (Table 3).

**Table 3. Genetic differentiation estimates among *Pinus sylvestris* adult and offspring samples from calcareous and siliceous soils at Sierra de Ayllón.**

| Sample | CAL-A | SIL-A | PLT-A | CAL-O | SIL-O |
|--------|-------|-------|-------|-------|-------|
| CAL-A | - | 0.157 | 0.135 | 0.124 | 0.175 |
| SIL-A | -0.00024 | - | 0.157 | **0.238** | 0.151 |
| PLT-A | 0.00004 | 0.0002 | - | 0.124 | 0.172 |
| CAL-O | -0.00046 | 0.0009 | -0.00018 | - | 0.185 |
| SIL-O | 0.00014 | -0.00008 | 0.00064 | 0.00036 | - |

CAL-A, SIL-A and PLT-A, adult samples from calcareous and siliceous natural populations and from nearby plantations, respectively; CAL-O and SIL-O, offspring samples from calcareous and siliceous natural populations; above diagonal: $D_{est}$, Jost's differentiation index; below diagonal: $F_{ST}$, AMOVA-based population differentiation index. Estimates in bold are significantly different from zero (after Bonferroni correction with global $\alpha = 0.05$).

The distribution of observed haplotypic counts provided further details on the high haplotypic diversity and moderate differentiation among groups (S1 Table). Out of a total of 247 different cpSSR haplotypes observed across all five adult and offspring sample categories, 162 (66%) of them were detected at most in one individual within each category, and the maximum observed haplotypic frequency was below 7%. Among the adult samples, there were a total of 207 different haplotypes, of which a similar number (about 29%) were specific to (i.e. only found in) either the natural stands or the plantations. The cumulative frequency (CF) of haplotypes specific to one of the three adult sample categories was 0.153, 0.153 and 0.190 for CAL-A, SIL-A and PLT-A, respectively.

Regarding offspring samples, recruits from the two natural stands carried a majority (CF = 0.878 and 0.830 for CAL-O and SIL-O, respectively) of haplotypes that were also detected among one or more of the adult sample categories. As a potential evidence of male gametic exchange across stands, a small proportion of the haplotypes observed among offspring from the calcareous natural stand were only detected among adults of the siliceous natural stand (CF = 0.020) or of the plantations (CF = 0.041). Similarly, offspring sampled within the siliceous stand carried a few haplotypes only observed among adults from the calcareous stand (CF = 0.032) or from the plantations (CF = 0.048).

## Gene flow estimates

The Bayesian estimates indicated substantial gene exchange between the two natural populations (Table 4). The estimated proportion of offspring of the calcareous population sired by fathers from the siliceous population was about 7.6%, while the estimated proportion of offspring of the siliceous population sired by fathers from the calcareous population was 21.5% (Table 4). Both estimates were significantly different from zero, although with wide credibility intervals. Gene flow estimates from the plantations into the natural populations were larger, with 64.4% and 42.3% of offspring in the calcareous and siliceous populations, respectively, sired by fathers from planted stands (Table 4).

## Simulation analysis of model performance

All MCMC runs converged without problems for all the simulated data sets. The obtained expected errors for migration estimates were very similar under the three alternative adult haplotypic frequency distributions assumed in the simulations (see S2 Table), so we focus here on the ones obtained assuming $nh$ = 247. Simulation results indicated that the Bayesian inference model produced male gametic gene flow estimates with generally low biases, assuming gene flow rates from 0% to 60% from the two natural and the planted populations (Fig 2 and S2 Table). Expected estimation errors were similar for gene flow rates into the calcareous and the siliceous populations. Without any gene flow, the model produced positive gene flow rate

**Table 4. Estimates of male gametic gene flow rates among *Pinus sylvestris* study populations.**

| RECIPIENT POPULATION | SOURCE POPULATION | | |
|---|---|---|---|
| | CAL | SIL | PLT |
| CAL | 0.254 (0.018–0.680) | 0.076 (0.004–0.275) | 0.644 (0.241–0.904) |
| SIL | 0.215 (0.013–0.598) | 0.342 (0.055–0.607) | 0.423 (0.157–0.701) |

CAL, calcareous natural population; SIL, siliceous natural populations; PLT, plantations. The estimates (and 95% credibility intervals) were obtained for naturally established recruits within the two natural populations. CAL-CAL and SIL-SIL values represent local dispersal rates.

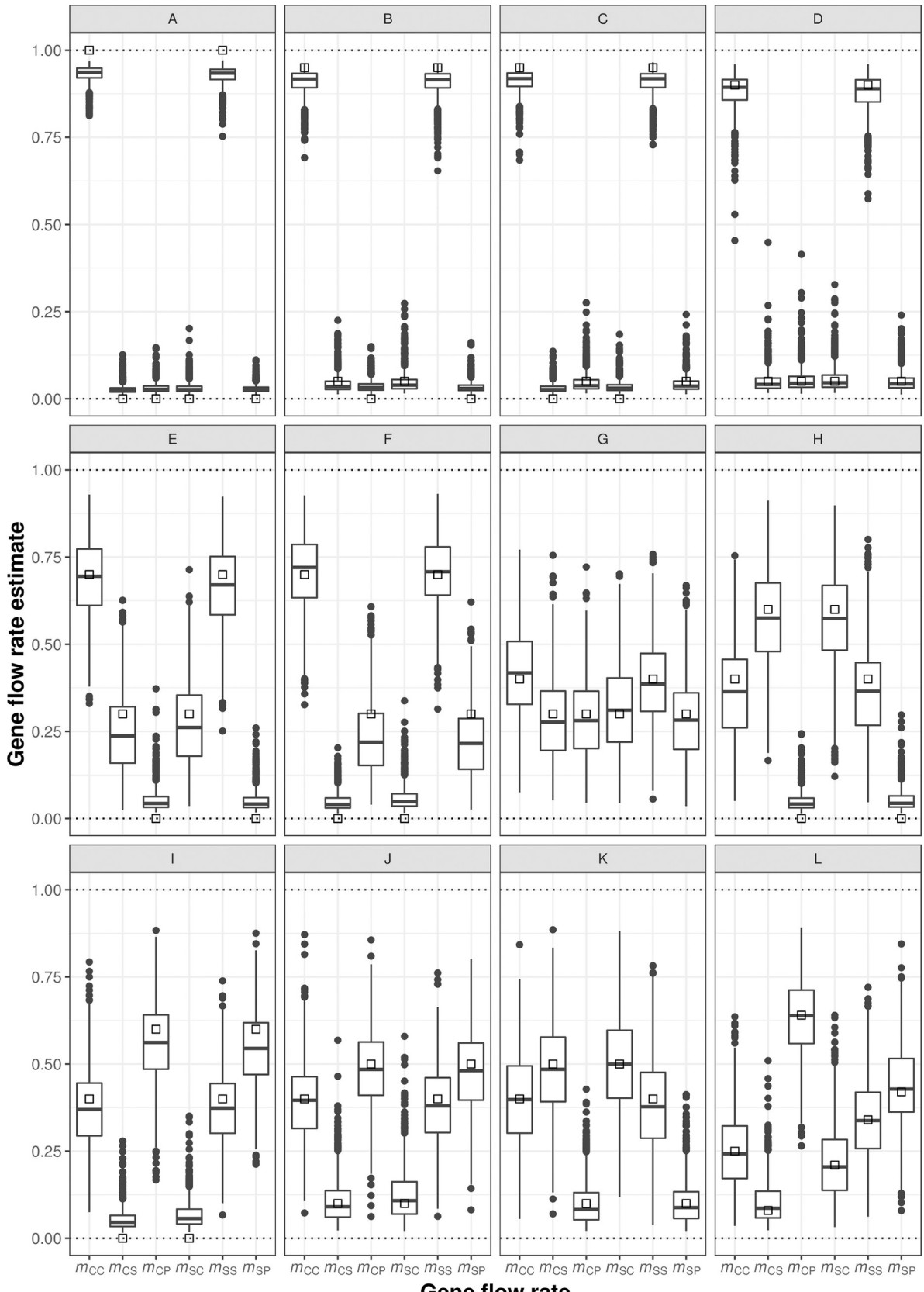

**Fig 2. Distribution of estimated male gametic gene flow rates in simulated gene flow scenarios.** Different simulated scenarios (A–L) of gene flow among the three *Pinus sylvestris* study populations were simulated. Each panel corresponds to a different simulated scenario with a different assumed set of six pairwise male gametic gene flow rates ($m$), with the first and second subindexes of $m$ indicating the recipient and source population, respectively, with codes $C$ (calcareous natural population), $S$ (siliceous natural population) and $P$ (plantations). The six little white squares within each panel indicate the assumed $m$ values in the corresponding scenario. Boxplots indicate the median, first and third quartiles and the 5th and 95th percentiles, with outliers plotted as black dots. The scenarios are described in Table 1. Results shown were obtained assuming a total of $nh$ = 247 haplotypes in the simulated populations.

estimates of about 3% from every external source population (Fig 2A, S2 Table). This residual positive bias for null gene flow rates tended to increase (up to maximum values of 5–8%) when gene flow from some populations remained null while being increasingly large from others (Fig 2B–2F; S2 Table). Biases decreased with increasing gene flow rates, and were minimal in scenarios with some gene flow from every source populations (e.g. Fig 2G, 2J–2L; S2 Table). The variance and the absolute RMSE increased, however, with increasing gene flow rates. Overall, the model properly discriminated unequal levels of gene flow from different source populations, including scenarios with gene flow between the two natural stands but not from the plantations, and the reverse (e.g. Fig 2E and 2F, 2H–2L). The last simulated scenario assumed gene flow values similar to the ones estimated empirically, the results showing (as for previous scenarios with non-null gene flow from every source population) rather small biases and moderate RMSE (between 0.06 and 0.12).

Simulation results also showed that estimates of uncertainty (i.e. credibility intervals) around gene flow rate estimates were generally close to nominal levels, except for simulated gene flow rates of zero (S2 Table). Specifically, 95% credibility intervals had expected coverage rates above the nominal 95% value for assumed gene flow rates of 0.05 and 0.10, and remained between 86% and 95% for assumed gene flow rates of 0.3 and 0.5. By contrast, because of residual positive estimation biases, the expected coverage rates of 95% credibility intervals dropped below 60% for assumed gene flow rates of zero, down to a minimum of 3% when estimating zero gene flow rates in scenarios with high (0.60) gene flow rates originating from other sources (S2 Table).

## Discussion

We presented a Bayesian approach based on paternally inherited haplotypic data that enables the estimation of recent gene flow rates among large tree populations. Using this approach, we found substantial gene flow rates (8 and 21%) among two natural *P. sylvestris*) populations growing on contrasting edaphic conditions six kilometers apart from each other. We also found that recent gene immigration from nearby plantations into the two natural populations was even greater (42 and 64%). Numerical simulation results suggested that the model can be expected to yield reasonably accurate estimates for our empirical data sets, except for slight to moderate positive biases in hypothetical scenarios without gene flow.

### Gene flow among large Scots pine populations

Using pollen traps and flowering phenology observations, [3] and [27] detected airborne transport of germinable pollen among Scots pine populations several hundred kilometers away. Similar studies for other wind-pollinated species [1] seem to confirm that airborne viable pollen clouds do frequently bridge the spatial gap between distant conspecific tree populations. It was also known that a variable proportion of pollen transported over long distances manages to overcome potential phenological barriers and outcompete sparse local pollen clouds when reaching relatively small recipient populations, enabling long-distance pollen gene flow into the latter [38]. However, we ignored whether the dilution effect produced by abundant local

pollen sources within large recipient populations [12] might minimize effective long-distance pollination. Results from the present study suggest that local pollen shedding within large tree populations does not preclude long-distance effective pollen immigration from other large populations, demonstrating for the first time recent male gametic gene flow among widespread wind-pollinated populations over a 10 km scale. Consistently with our initial hypothesis of high recent gene flow among the study populations, the estimated frequency of recruits sired by non-local males during the last couple of decades was high in the two study stands, indicating widespread pollen movement across the study region and absence of flowering phenology barriers.

Several hypotheses could explain the high siring success estimated for external populations, despite the proximity advantage of local pollen donors within the recipient populations. First, effective pollen immigration might have been enhanced if non-local pollen grains had arrived at least partly when local female strobili were receptive but local male strobili were still immature, a phenological pattern that is well documented in Scots pine [3, 39]. Second, the large male gametic immigration rates could have resulted from gene flow from widespread unsampled external sources extending beyond the study area (scattered planted stands extend 30–40 km west and south of the sampling area shown in Fig 1), as the used method tends to perceive gene flow from unsampled external sources as gene flow from sampled external sources [20]. If this potential explanation were true, it would only tend to increase the inferred scale of among-population gene flow in the study system. A third explanation could be that immigration estimates were positively biased, but our simulation results indicate that this bias should be minimal when immigration rate estimates are as large as the ones we obtained.

The uncertainty about biological and physical factors determining the observed distribution of among-population gene flow rates does not undermine the high overall level of male gametic gene exchange observed among the study populations. We assume that such exchange has been mediated mostly by pollen flow, even if seed flow (which carries both male and female gametes), though more unlikely, cannot be discarded. Our empirical confirmation of Scots pine capacity for effective gene exchange among large populations supports the role of gene flow as a homogenizing evolutionary force contributing to the low molecular genetic differentiation among the species populations across its broad distribution range [26]. This direct empirical confirmation of effective contemporary gene flow among large distant populations is relevant, because gene flow is notoriously difficult to distinguish retrospectively from shared ancestral polymorphism as a potential factor contributing to weak population differentiation among big populations (e.g. [40]). Our results also indicate the potential for reproductive and genetic connectivity among large fragmented stands of wind-pollinated tree species, as well as the potential for exotic gene flow from allochthonous plantations into natural conspecific populations.

## Adaptive genetic divergence under high gene flow?

Our results draw attention to a potential scenario of adaptive genetic divergence under high gene flow. To the extent that historical gene flow between the two natural Scots pine stands had been as high as estimated for the recent reference time period (i.e. the last 20 years), one would in principle expect it to have hampered adaptive genetic differentiation. However, common garden experiments have revealed differences between the two populations in seedling emergence and early seedling growth, phenology and survival of similar magnitude than pairs of Iberian provenances hundreds of kilometers away [22]. These phenotypic differences, albeit attenuated, persisted five years after transplantation from the nursery to the common gardens [29].

Assuming our gene flow estimates are not positively biased, several hypotheses could reconcile observed quantitative genetic divergence with the high gene flow estimates in this study. Although available data do not allow assessing whether the two populations are locally adapted, a first hypothesis is that the sharp spatial differences in edaphic conditions are inducing divergent selection of such intensity to enable short-scale local adaptation despite high gene flow [5]. Scots pine is usually considered soil-indifferent, as it is found growing on both calcareous and siliceous soils, but tolerance to different substrates does not necessarily imply absence of substrate-driven selective pressures. It is known for instance that Scots pine has the potential to genetically evolve in response to different mycorrhizal fungi communities [41], while local adaptation to substrate has been demonstrated for other pine species [42]. Several genomic architecture models could enable the maintenance of adaptive genetic variation for polygenic traits under high gene flow, such as the shift toward fewer larger-effect loci or groups of small-effect loci in tight linkage (see review in [43]). A second hypothesis is that the maintenance of adaptive genetic divergence is being facilitated because lifetime selection ultimately reduces effective gene flow rates among the two study populations, relative to the high values estimated for our young recruit cohort. Most of our sample corresponded to young immature individuals, and reduced immigrant fitness at later life stages cannot be discarded. In line with this hypothesis, Wright [42] found evidences of local adaptation to serpentine soils only at ages older than 30 years in *Pinus ponderosa*, with a very low correspondence between relative early family performance and relative family performance after 36 years of growth. The third hypothesis is that the two study populations have not reached migration-selection balance, having genetically diverged in greater spatial and reproductive isolation, and coming into their present-day relative proximity only recently. It is in fact known from macrofossil records that currently nearby Scots pine mountain populations in the region were in some cases separated during cold-stage expansions into warmer lowlands [44], and the two study populations in particular show a relatively distant genealogical relationship that is consistent with greater historical isolation [45]. Finally, an additional hypothesis is that the level of genetic divergence between the two study populations reported in the literature might have been inflated by maternal effects. The two populations grow in contrasting habitats, and the maternal environment is known to affect especially early offspring performance in plants, independently of genetic heredity [46]. If common garden experiments are established from seeds collected directly from maternal plants, as is generally the case in long-living trees, it is advisable to partially account for potential environmental maternal effects in early offspring performance by including seed weight as covariate [46], which was not the case in the two studies mentioned above on the focal populations.

Overall, we believe that a combination of these hypotheses might explain the apparent joint presence of high gene flow and strong adaptive genetic differentiation: divergent selection between habitats, decreasing migrant fitness with age, migration-selection disequilibrium, and environmental maternal effects increasing early phenotypic differentiation measures in common gardens.

## Methodological considerations and perspectives

Measuring recent gene flow among large, genetically diverse and lowly differentiated populations confronts identifiability and sampling issues. Low differentiation tends to make migrants undetectable, while large and diverse populations imply larger reference samples for accurate estimation of haplotypic frequencies. High genetic diversity does also complicate the estimation of descriptive population differentiation measures such as $F_{ST}$, which approaches zero when within-population diversity is high even if populations are completely differentiated

[35]. This is illustrated by our results, where the high haplotypic diversity (66% of haplotypes observed at most once within each population sample and a maximum population haplotype frequency of 7%) translates into virtually null $F_{ST}$ estimates but larger values of Jost's $D_{est}$ measure of population dissimilarity (Table 3), which is independent of within-population diversity. Our simulation analysis seems to suggest that $D_{est}$ is a better indicator than $F_{ST}$ of statistical identifiability of recent gene flow rates inference from genotypic data, as the expected accuracy of the model for our data set is better than expected if differentiation was actually zero. The anticipation of high population genetic diversity led us to invest substantial resources in sampling, with about 1 200 individuals collected and genotyped, which could explain the acceptable expected accuracy of the model. However, still larger samples would probably have further reduced gene flow rate estimation errors, in particular the residual positive bias expected without gene flow. Although this residual bias does not seem very problematic in our particular study system (with high gene flow rates), it highlights the big challenge of estimating low-frequency gene flow among lowly differentiated populations.

Our Bayesian approach estimates recent migration rates incorporating uncertainty in population haplotypic frequency estimation, under the prior assumption that any haplotype detected in any of the population samples is also present in the other populations, and updating this prior with observed haplotypic counts [37]. We tried to avoid in this way potential biases derived from undetected low-frequency haplotypes in adult samples. On the other hand, the presented model does not incorporate genotyping errors. Unaccounted-for mistaken alleles can potentially increase the errors of pollen migration rate estimates based on genetic assignment methods using diploid biparentally inherited markers [21]. For the different case of uniparentally inherited haplotypic data, it would be relevant to test the effect on male gametic migration rate estimates of microsatellite step-wise length scoring errors, one of the most common sources of error for these markers [47]. Our unpublished results show that migration rate estimates obtained with our model are insensitive to this kind of scoring errors for our particular empirical population haplotypic frequency distribution, irrespective of assumed gene flow levels. An unpublished extension of our inference model that jointly estimates mistyping rates produced virtually identical lowly biased migration rate estimates as the model without mistyping, on simulated data with our empirical haplotype frequency distribution, while severely underestimating mistyping rates. The negligible effect of mistyping on migration rate estimation, and the difficulty to estimate mistyping rates, likely result both from the very high number of low-frequency haplotypes and the fact that most haplotype pairs differ in several of the six considered cpSSR regions (S1 Table), which make mistyping difficult to detect but also renders systematically biased population assignments very unlikely.

To conclude, beyond increasing population sample sizes and extending the spatial scale of analysis for more accurate gene flow characterization, future empirical studies could obtain estimates of population flowering phenology overlap (as in [3, 11]). These data could be combined in the model with haplotypic likelihoods to explore the relative effects of spatial versus phenological distances on among-population gene flow intensity and direction.

## Supporting information

**S1 Appendix. PCR protocols.**
(DOCX)

**S2 Appendix. Details of the MCMC algorithm.**
(DOCX)

**S3 Appendix. Monte Carlo analysis of method performance.**
(DOCX)

**S1 Table. Observed haplotype counts and assumed frequencies for simulations.**
(XLSX)

**S2 Table. Expected errors of gene flow rate estimates.**
(XLSX)

## Acknowledgments

We thank Rodrigo Pulido (INIA-CIFOR, Madrid, Spain) for helping with field work, Carmen García (INIA-CIFOR) for laboratory assistance, and Jesús Martínez-Fernández (INIA-CIFOR) for help in producing Fig 1. We also thank the staff at the Dirección Provincial de Agricultura, Medio Ambiente y Desarrollo Rural en Guadalajara (Dirección General de Política Forestal y Espacios Naturales) for field work authorization.

## Author Contributions

**Conceptualization:** Juan José Robledo-Arnuncio.

**Data curation:** Azucena Jiménez-Ramírez, Delphine Grivet, Juan José Robledo-Arnuncio.

**Formal analysis:** Azucena Jiménez-Ramírez, Juan José Robledo-Arnuncio.

**Funding acquisition:** Delphine Grivet, Juan José Robledo-Arnuncio.

**Investigation:** Azucena Jiménez-Ramírez, Juan José Robledo-Arnuncio.

**Methodology:** Juan José Robledo-Arnuncio.

**Project administration:** Delphine Grivet, Juan José Robledo-Arnuncio.

**Software:** Juan José Robledo-Arnuncio.

**Supervision:** Delphine Grivet, Juan José Robledo-Arnuncio.

**Validation:** Juan José Robledo-Arnuncio.

**Writing – original draft:** Azucena Jiménez-Ramírez.

**Writing – review & editing:** Delphine Grivet, Juan José Robledo-Arnuncio.

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
