## [Decision Letter · Decision Letter 0]

8 May 2021

PONE-D-21-09948

Measuring recent effective gene flow among large populations in *Pinus sylvestris*: local pollen shedding does not preclude substantial long-distance pollen immigration

PLOS ONE

Dear Dr. Robledo-Arnuncio,

Thank you for submitting your manuscript to PLOS ONE. After careful consideration, we feel that it has merit but does not fully meet PLOS ONE’s publication criteria as it currently stands. Therefore, we invite you to submit a revised version of the manuscript that addresses the points raised during the review process.

The manuscript has been met with great interest on the side of the review team. Before acceptance, I would like to see a reply to the comments and many suggestions, as detailed further below. Overall, to me this is someway between a 'minor' and 'major' revision (the reviewers may want to see the revised version, out of their pure interest in the first place).

We look forward to receiving your revised manuscript.

Kind regards,

Berthold Heinze

Academic Editor

PLOS ONE

Additional Editor Comments:

The manuscript has been reviewed by three colleagues who seem to be very interested in the approach and the results. They have all made suggestions that may improve the study and its presentation and interpretation, many thanks for their efforts! I am very much looking forward to a reply by the authors, which I expect will be a very valuable discussion of these points. I do not think that every suggestion must be followed, but a few of them are certainly worth a trial (e.g. the smplification of haplotypes). I leave it to the authors to choose among those and argue how best to improve the manuscript, and I hope it will be convincing to us.

Journal Requirements:

2. We note that Figure 1 in your submission contain map images which may be copyrighted. All PLOS content is published under the Creative Commons Attribution License (CC BY 4.0), which means that the manuscript, images, and Supporting Information files will be freely available online, and any third party is permitted to access, download, copy, distribute, and use these materials in any way, even commercially, with proper attribution. For these reasons, we cannot publish previously copyrighted maps or satellite images created using proprietary data, such as Google software (Google Maps, Street View, and Earth). For more information, see our copyright guidelines: http://journals.plos.org/plosone/s/licenses-and-copyright.

2.1.    You may seek permission from the original copyright holder of Figure 1 to publish the content specifically under the CC BY 4.0 license. 

2.2.    If you are unable to obtain permission from the original copyright holder to publish these figures under the CC BY 4.0 license or if the copyright holder’s requirements are incompatible with the CC BY 4.0 license, please either i) remove the figure or ii) supply a replacement figure that complies with the CC BY 4.0 license. Please check copyright information on all replacement figures and update the figure caption with source information. If applicable, please specify in the figure caption text when a figure is similar but not identical to the original image and is therefore for illustrative purposes only.

Reviewers' comments:

Reviewer's Responses to Questions

**Comments to the Author**

1. Is the manuscript technically sound, and do the data support the conclusions?

Reviewer #1: Yes

Reviewer #2: Yes

Reviewer #3: Partly

2. Has the statistical analysis been performed appropriately and rigorously? 

Reviewer #1: Yes

Reviewer #2: Yes

Reviewer #3: Yes

3. Have the authors made all data underlying the findings in their manuscript fully available?

Reviewer #1: Yes

Reviewer #2: Yes

Reviewer #3: Yes

4. Is the manuscript presented in an intelligible fashion and written in standard English?

Reviewer #1: Yes

Reviewer #2: Yes

Reviewer #3: Yes

5. Review Comments to the Author

Reviewer #1: Long-distance pollen dispersal indeed is a focus of several recent studies. However, solution how to estimate it are generally lacking. This manuscript provides one of the solutions that might be applied to conifers because the authors used paternally inherited markers (cpSSRs). Any attempt to bring us closer to solving the dilemma about the extent of long-distance pollen-mediated gene flow deserves attention.

I generally find the manuscript well designed. The sample size and markers used are fine. Novel aspects include the Bayesian estimation method, supported by relevant simulations. The general conclusion supports our expectations that pollen-mediated gene flow in trees such as Scots pine might be extensive and may significantly contribute to newly generated offspring cohorts.

I fully support this manuscript for publication in PLOS ONE. However, I have some comments and suggestions that might appear useful for the authors to improve the manuscript.

The authors mention in several places that haplotype frequency estimation uncertainty has little effect on the precision of the immigration levels. This problem is only briefly mentioned in methods that in one of the simulations, all populations have equifrequent haplotypes (line 301). The frequency distribution of haplotypes, especially when they are multiple, as in Scots pine, raises some questions about the sample size to approximate actual frequencies representing particular populations. With multiple haplotypes with very low frequency, the sampling variance for the frequencies of low-frequency haplotypes are large, and the ability to include a particular haplotype in the sample is small. This at least should be mentioned in the manuscript.

A variance of cpSSR haplotypes also depends on the variance of the fecundity of adults in the ‘source’ populations. I wonder if this model is sensitive to the variation of haplotype frequency due to the variable male fecundities in the source populations?

What is the age of plantations? Did they reach the full capacity of pollen production since establishment?

What is the approximate proportion of native populations and plantations. Based on Fig 1, plantations demonstrate a low proportion of forested area as compared to natural populations.

The origin of plantations might be of some importance. Their relatedness to one or both natural stands may cause an overestimation of immigration levels for plantations. If those plantations originate from seeds derived from natural local stands, then the mixture of both natural stands may likely form a likely source of populations. This is probably not a very sensible idea, but I would try to generate an artificial (in silico) population generated as a random mixture of haplotypes from the two natural stands, then mimic this mixture population as a source of pollen (instead of actual plantations), and see how this could influence estimation process.

Line 174, based on fig 1, it cannot be said that sampling of adults and offspring was random. It was not random but done in kind of lines/transects/trails ???) both in natural stands and plantations. It could be random in respect to haplotypes, because I have never seen any fine-scale spatial genetic structure in any conifer based on cpSSRs.

Line 177, the distance between sampled adults may not be so essential to avoid relatedness based on cpDNA markers in conifers.

Are there any additional scots pine stands in the area not covered by figure 1, that could be the source of immigrant pollen? This is important for discussion.

Simulations were fine, but they all assumed that the frequency distribution of haplotype frequency was precisely estimated. I wonder, what would be the effect if the frequency of haplotypes would be estimated with bias?

Table 3 Suggest putting ‘-‘ instead of 0.000 on the diagonal.

Line 482, the statement ‘we found a large proportion of recruits sired by non-local males’ does not seem appropriate because this sounds like the evidence-based on paternity analysis (genetic exclusion). I suggest talking about this in terms of ‘probability’ of non-local siring.

Line 491, Scots pine is known to be protogynous at individual and population levels, which means that female flowers are exposed to pollen before local pollen shedding, contributing to increased pollen immigration. There is a vast amount of references that document this phenomenon.

Several important papers touch on the problem of extensive pollen-mediated gene flow in Scots pine and conifers and might be considered as references. Some of long distance pollen dispersal has been presented in:

Lindgren, D., Paule, L., Xihuan, S., Yazdani, R., Segerström, U., Wallin, J.-E., & Lejdebro, M. L. (1995). Can viable pollen carry Scots pine genes over long distances? Grana, 34(1), 64-69.

Nilsson, J. E. (1995). Genetic variation in the natural pollen cloud of Pinus sylvestris: a study based on progeny testing. Scandinavian Journal of Forest Research, 10(1-4), 140-148.

Burczyk, J., Sandurska, E., Lewandowski, A,. (2019). Patterns of Effective Pollen Dispersal in Larch: Linking Levels of Background Pollination with Pollen Dispersal Kernels. Forests, 10(12), 1139.

Line 519, wrong citation, the cited paper referred to the problem of gene flow between species in terms of hybridization and should not be confused with gene flow among populations of the same species. Try to find a better reference. Indeed, in the case of low to none genetic differentiation among populations measuring the levels of contemporary gene flow based on differentiation is inefficient.

Line 527-530, it is not only gene flow that determine differentiation but also (rather) the intensity of selection of adaptive traits. The intensity od selection in trees is high, because of massive production of offspring (in climax conditions on average one adult is replaced by one offspring, but each individual may produce thousands to millions of offspring over a lifetime).

Line 613-615, but there are methods to deal with genotyping errors and nulls in nuclear markers, see Chybicki 2018 (Chybicki, I. J. (2018). NMpi-improved re-implementation of NM+, a software for estimating gene dispersal and mating patterns. Mol Ecol Resour, 18(1), 159-168. doi:10.1111/1755-0998.12710)

Line 617, where the effect of step-wise length scoring errors simulations results are presented? Cannot find it.

Lines 631-642, some of the future perspectives of research cover issues not related directly to the studied phenomenon – gene flow, but focus on the studying of patterns and evidences of genetic local adaptation.

Reviewer #2: The manuscript is focused on the estimation of pollen-mediated gene flow in a wind-pollinated conifer (Pinus sylvestris) at the southern margin of its distribution range in Spain. However, the focus is divided into two components: the biological questions related to long-distance pollen dispersal and the methodological improvements made on earlier statistical methods developed by some of the authors. The range of long-distance gene dispersal remains a largely unanswered question, mostly due to how difficult such dispersal events are detected. In wind-pollinated trees, pollen-mediated gene flow is mostly inferred using paternity exclusion which is ineffective in determining distances outside the local population. The methods used in this study rely on the opposite approach, where pollen gametes are fractionally assigned to different, discretely defined gene pools, each of which representing a different geographic location. Such an approach is quite well known and long-used by the community and resembles ideas implemented in the BayesAss software, but adjusted specifically to plants.

Regarding the biological merit, the study seems to be inspired by the phenotypic divergence between pine populations, which provides a relevant basis for the hypothesis about migration and, possibly, divergent selection. The latter is mentioned as a hypothetical factor determining differences in early growth and phenology between the two population types, especially if recurrent migration takes place in both directions.

Generally, and especially taking into account the criteria of the PLOS One, I believe the manuscript is suitable for publication. The hypothesis is clear and sound. The methods are correct. Even if the biological findings are not truly novel (Scots pine is long known for its high pollen dispersal potential), they offer some novelty, especially in the part related to methodological improvements. However, before the final decision, I recommend some revisions according to the following suggestions (provided point by point):

L178: It is generally acknowledged that a presence of relatives in a sample influences estimates of parental (say ancestral) allele frequencies. Such “ancestral” frequencies are typically needed to estimate unbiased genetic indices related to the probability of identity by descent in the current generation. But, as long as I understand, the estimation method used in this study does not involve such indices, but it is focused on assigning next-generation (progeny) genotypes to the current-generation propagule sources. Such an approach relies on the current generation allele frequencies, and not the previous generation (ancestral) ones. Therefore, intentional avoiding relatives can introduce ascertainment bias to estimated gene pools of current seed and pollen sources. Given the species' dispersal potential, family structure is likely to be very low so the mentioned bias has likely a negligible impact on the presented results. On the other hand, and seeking generality, in the case of strong family structure, the bias may no longer be ignorable. In my opinion, a random sample from a source population should be used. I would like to know opinion of the authors on this issue.

L222: lower-case J should be used in “… from population J”

L255: the use of “uninformative” for uniform priors is not generally accepted because such prior distributions still introduce some information. Please use “weakly informative” instead.

L265: the simulations were run using the posterior haplotype frequencies (L299) obtained under (I guess) the assumption of the uniform priors. It is then not surprising that the other priors used to model simulated data gave worse results. By the way, concentration parameters (alphas) of 1/K or 1/J in the Dirichlet distribution represent the so-called Jeffreys prior which is considered to be an uninformative prior under some assumptions.

L268: maybe you mean “Chain rule”

L278: but it does not reduce the ascertainment bias due to non-random sampling from source populations

L281: “than” should read “to”

Table 3: Using a bolded font instead of italicized one should help “extracting” the significant value out.

L456: Given so wide credible intervals for migration vectors, I am not convinced that gene flow is generally asymmetric. To conclude that, the fit (or probability) of the estimation model needs be compared vs. the null model of symmetric gene flow.

L459: remove double “that”

L520: but the potential can depend on the species absolute number, i.e. the number of trees shedding pollen. In the case of substantial demographic depression, when the number of trees per forest patch is of order of tens or less, the total amount of pollen grains can be insufficient to effectively pollinate female strobili in another patch (the effect of pollen dilution).

L537: The estimation of pollen migration is based exclusively on chloroplast genetic data. In order to extrapolate the observed migration patterns to the nuclear genome, the assumption of cyto-nuclear equilibrium must hold that would imply equivalence of divergence rates (up to the ploidy factor) between cp and nuclear genomes. However, divergence rates can differ due to selection affecting differently the two genomes. Assuming that the phenotypic divergence observed in the study populations has a genetic basis, it is mostly related to the selection on the nuclear genes. Then, cp and nuclear genomes may not be equivalent in so-called effective (i.e. after selection) gene flow patterns. Under such circumstances, neutral cpDNA can over-estimate the effective gene flow. I would like to know opinion of the authors about that.

L602: Rather than affecting “errors”, large samples will increase precision of estimates. However, increased precision is only one side of the coin. If accuracy remains unaffected, higher precision will lead to false positives, such as significant (non-zero) migration in the case of no migration. This may limit the utility of the proposed method to non-zero migration cases only. Here, again, the lack of formal test of the null hypothesis (no migration) decreases the value of the proposed method.

S2 Appendix:

L49-51: If 260,000 iterations are used in total, including 10,000 discarded iterations for burn-in, then the thinning of 25 samples gives 10,000 samples from the posterior, and not 1,000 (as said in the text).

Reviewer #3: Review of the manuscript:

Measuring recent effective gene flow among large populations in Pinus sylvestris: local pollen shedding does not preclude substantial long-distance pollen immigration

The manuscript addresses the important topic of effective long distance gene flow by pollen among tree populations. Accurate estimations of this long distance pollen immigration is quite challenging for large wind pollinated tree populations. The authors observed high and asymmetrical pollen immigration rates in two natural Pinus sylvestris stands and discussed how this could be explained despite adaptive differences observed in a common garden experiment.

The main merit of the paper is the presented Bayesian approach to simultaneously estimate haplotype frequencies and immigration rates. The weak point- which has also been addressed by the authors- is the high variation at the chloroplast microsatellites that lead to an extreme high number of haplotypes (line 361 ff.). Even with the high sample sizes presented in the paper all estimates of haplotype frequencies remained imprecise. As a consequence, nearly all pairwise comparison of genetic differences in haplotype frequencies are statistically not significant (line 323 ff., table 3) and all estimates of pollen migration rates had very large credibility intervals (line 389).

I suggest to apply a “rarefaction” approach to check the robustness of the estimated pollen immigration rates. With combinations of less than the six microsatellites the number of haplotypes will decrease and the confidence intervals of the frequency estimations will all get narrower. All measures could be repeated with different numbers of combined microsatellites.

Beside this main point I have only a few smaller remarks that might help to improve the paper:

Line 180 ff.: Why offspring and not seeds have been used? How was the sampling design of the offspring?

Line 482: Discuss the impact of the age of offspring: At least some of them might be older than a couple of years. And the haplotype frequencies in the adult population might have changed since then.

Line 537 ff.: One additional and simple explanation for the discrepancy of local adaptation and high effective pollen immigration would be that the pollen immigration has been overestimated. There are other population genetic and demographic parameters that could create differences between haplotype frequencies in adults and offspring of the same stand: e.g. drift effect due fertility variation of adults.

6. PLOS authors have the option to publish the peer review history of their article (what does this mean?). If published, this will include your full peer review and any attached files.

Reviewer #1: No

Reviewer #2: No

Reviewer #3: No

---

## [Author Response · Author response to Decision Letter 0]

15 Jun 2021

The response to the editor and reviewers comments is included in the attached "response_letter.pdf" file.

---

## [Decision Letter · Decision Letter 1]

26 Jul 2021

Measuring recent effective gene flow among large populations in *Pinus sylvestris*: local pollen shedding does not preclude substantial long-distance pollen immigration

PONE-D-21-09948R1

Dear Dr. Robledo-Arnuncio,

We’re pleased to inform you that your manuscript has been judged scientifically suitable for publication and will be formally accepted for publication once it meets all outstanding technical requirements.

Kind regards,

Berthold Heinze

Section Editor

PLOS ONE

Additional Editor Comments (optional):

The reviewers are happy with the changes and explanations, and so am I. I think this comes out as a very interesting approach, and although it has some caveats and specific circumstances, it may serve to inform others in this important topic.

Reviewers' comments:

Reviewer's Responses to Questions

**Comments to the Author**

1. If the authors have adequately addressed your comments raised in a previous round of review and you feel that this manuscript is now acceptable for publication, you may indicate that here to bypass the “Comments to the Author” section, enter your conflict of interest statement in the “Confidential to Editor” section, and submit your "Accept" recommendation.

Reviewer #1: All comments have been addressed

Reviewer #2: All comments have been addressed

2. Is the manuscript technically sound, and do the data support the conclusions?

Reviewer #1: Yes

Reviewer #2: Yes

3. Has the statistical analysis been performed appropriately and rigorously? 

Reviewer #1: Yes

Reviewer #2: Yes

4. Have the authors made all data underlying the findings in their manuscript fully available?

Reviewer #1: Yes

Reviewer #2: Yes

5. Is the manuscript presented in an intelligible fashion and written in standard English?

Reviewer #1: Yes

Reviewer #2: Yes

6. Review Comments to the Author

Reviewer #1: (No Response)

Reviewer #2: Dear Authors, thank you for carefully addressing my comments on the initial submission. I am pleased to say I am satisfied with your replies. Regarding the comment about possible differences between cp and nu(clear) genomes in effective gene flow rates (original L537) raised in the original review, I wanted to say that phenotypic selection acting on some tree traits may limit the effective gene flow at the nuDNA level. However, at the cpDNA level, such an effect may remain invisible so that estimates of current gene flow shown in this study may not incorporate this effect leading to a sort of discrepancy between high gene flow estimates and high phenotypic divergence. This issue, however, affects neither the overall impact of the paper nor the conclusions.

7. PLOS authors have the option to publish the peer review history of their article (what does this mean?). If published, this will include your full peer review and any attached files.

Reviewer #1: No

Reviewer #2: No

---

## [Editor Report · Acceptance letter]

5 Aug 2021

PONE-D-21-09948R1 

Measuring recent effective gene flow among large populations in Pinus sylvestris: local pollen shedding does not preclude substantial long-distance pollen immigration 

Dear Dr. Robledo-Arnuncio:

I'm pleased to inform you that your manuscript has been deemed suitable for publication in PLOS ONE. Congratulations! Your manuscript is now with our production department. 

Kind regards, 

on behalf of

Dr. Berthold Heinze 

Section Editor

PLOS ONE